# EEG-Based Music Emotion Prediction Using Supervised Feature Extraction for MIDI Generation

**DOI:** 10.3390/s25051471

**Published:** 2025-02-27

**Authors:** Oscar Gomez-Morales, Hernan Perez-Nastar, Andrés Marino Álvarez-Meza, Héctor Torres-Cardona, Germán Castellanos-Dominguez

**Affiliations:** 1Faculty of Systems and Telecommunications, Universidad Estatal Península de Santa Elena, Avda. La Libertad, Santa Elena 7047, Ecuador; oscargomez@upse.edu.ec; 2Signal Processing and Recognition Group, Universidad Nacional de Colombia, Manizales 170003, Colombia; amalvarezme@unal.edu.co (A.M.Á.-M.); cgcastellanosd@unal.edu.co (G.C.-D.); 3Transmedia Research Center, Universidad de Caldas, Manizales 170003, Colombia; hector.torres_c@ucaldas.edu.co

**Keywords:** EEG, music emotion recognition, piano-roll algorithm, kernel methods

## Abstract

Advancements in music emotion prediction are driving AI-driven algorithmic composition, enabling the generation of complex melodies. However, bridging neural and auditory domains remains challenging due to the semantic gap between brain-derived low-level features and high-level musical concepts, making alignment computationally demanding. This study proposes a deep learning framework for generating MIDI sequences aligned with labeled emotion predictions through supervised feature extraction from neural and auditory domains. EEGNet is employed to process neural data, while an autoencoder-based piano algorithm handles auditory data. To address modality heterogeneity, Centered Kernel Alignment is incorporated to enhance the separation of emotional states. Furthermore, regression between feature domains is applied to reduce intra-subject variability in extracted Electroencephalography (EEG) patterns, followed by the clustering of latent auditory representations into denser partitions to improve MIDI reconstruction quality. Using musical metrics, evaluation on real-world data shows that the proposed approach improves emotion classification (namely, between arousal and valence) and the system’s ability to produce MIDI sequences that better preserve temporal alignment, tonal consistency, and structural integrity. Subject-specific analysis reveals that subjects with stronger imagery paradigms produced higher-quality MIDI outputs, as their neural patterns aligned more closely with the training data. In contrast, subjects with weaker performance exhibited auditory data that were less consistent.

## 1. Introduction

Music-generation algorithms have advanced significantly in recent years, enabling the creation of innovative and realistic compositions with a wide range of applications, including the generation of multimedia music for video games [1], movies and television shows [2], developing educational tools for teaching music [3], serving as a relevant therapeutic strategy for helping people with neurological disorders and children with mild learning disabilities [4,5], facial emotion recognition for mobile devices [6], among others [7]. However, bridging neural and auditory domains poses challenges, such as defining musical behaviors, linking neurophysiology to musical tasks, and addressing cognitive capacity limitations [8]. The main challenge is linking neural activity to musical perception and cognition [9]. Music engages multiple brain regions, requiring advanced analytical methods to decode its distributed neural patterns [10,11]. Brain–computer interface (BCI) devices, which utilize EEG to decode brain activity, are widely used for music interaction and control due to their non-invasiveness, portability, affordability, high temporal resolution [12,13], and ability to integrate multimodal sensors [14,15]. BCI systems are rapidly evolving as multisensory machine learning tools, enhancing mental and motor imagery cognitive functions that underpin creativity, memory, and planning [16,17]. Nonetheless, existing EEG-based emotional music-generation systems face key limitations. First, functional auditory cortical representations rely on hand-crafted low- and high-level features, often introducing subjective biases that constrain the musical experience [18,19]. Alternatively, the use of third-party software in music generation can result in a loss of control over the consistency between generated music and intended emotional expressions [20]. Therefore, a controllable music-generation framework is essential to ensure accurate correspondence with emotions inferred from EEG signals.

Although advancements in human–computer interaction highlight the importance of emotion recognition for system interpretation, EEG-based music generation faces additional restrictions such as signal noise, artifacts, and environmental interference [21]. In addition, capturing musical emotion is hindered by issues related to dataset quality, annotation consistency, and model generalization [22]. Therefore, rigorous data cleaning and preprocessing are essential for extracting reliable insights [23]. Another restriction is that EEG-based music cognition research is limited by single-modality neuroimaging, hindering the precise identification of brain regions and temporal sequences in musical processing [24]. Enhancing spatial sampling can be achieved by increasing electrode count or integrating EEG’s temporal resolution with other neuroimaging techniques via multimodal fusion for improved emotion recognition [25]. However, both methods are often constrained by high costs and limited portability [26]. The brain’s nonlinear dynamics challenge traditional EEG BCI analysis, which relies on simple features and may miss complex neural patterns [27]. To address this, data-driven deep learning (DL) models are increasingly used for emotion recognition from natural music stimuli [28,29,30]. However, training deep learning models on raw EEG data for music-related brain activity is challenged by high dimensionality, complex EEG dynamics [31,32], the need for large labeled datasets to prevent overfitting [33,34], the absence of long-term musical structures [35], and the lack of standardization in genres, styles, and moods [36], among others. Generally, DL-based music generation can be refined in terms of both neurophysiological models for emotion perception and musical structures, highlighting key areas for further cognitive and neurophysiological research [9]. Thus, recent advancements often adopt a hierarchical structure with multiple large models dedicated to semantic, coarse acoustic, and fine acoustic modeling [37,38,39]. At the same time, various approaches, such as emotion-conditioned models, integrate emotional cues to enhance the expressiveness and resonance of AI-generated music [40]. While these models improve the ability of AI-generated music to convey intended emotions—enhancing user experience and creative potential—they face challenges, including the subjective nature of emotion evaluation (which is highly context-dependent and difficult to model accurately) and the scarcity of high-quality datasets necessary for generating diverse and precise musical outputs [10]. However, the main limitation of incorporating training frameworks with more complex context models lies in generating music in real time while adapting to physiological signals. This requires efficient models and hardware, making implementation challenging due to the high computational costs, which can render real-time generation impractical [41].

Nonetheless, integrating DL models with EEG BCI devices imposes additional constraints, particularly in capturing fine-grained temporal and spatial neural patterns [42,43], as well as addressing individual variability, which hinders model generalization for controllable deep data generation [44,45]. Another key challenge in BCI-based music cognition research is linking features extracted from neural responses to musical concepts, emphasizing the need for clear associations between extracted neural features and high-level musical structures [46,47]. To address these challenges, this study introduces a novel DL framework for generating MIDI sequences that align with labeled emotion predictions using supervised feature extraction from the neural and auditory domains. Specifically, we employ the EEGNet model for neural data and an autoencoder-based piano algorithm for auditory data. Due to the heterogeneity between these high-dimensional modalities, we incorporate Centered Kernel Alignment to provide the largest separation between emotional states. To further enhance MIDI reconstruction quality, we propose a regression between both feature domains to minimize dispersion caused by the high intra-subject variability of EEG patterns. This is achieved by clustering the latent representations obtained through CKA for MIDI into denser partitions using the *k*-NN algorithm. Using musical metrics, the evaluation of real-world data shows that the proposed approach improves emotion classification (namely, between arousal and valence) and the system’s ability to produce MIDI sequences that better preserve temporal alignment, tonal consistency, and structural integrity. Regarding inter-subject EEG variability, the validation results indicate that the higher the accuracy of emotion classification labels, the more consistent the quality of auditory outputs. Consequently, the performance of the proposed DL framework may decline significantly for subjects with limited proficiency in BCI usage.

## 2. Methods

### 2.1. Auditory Feature Extraction

#### 2.1.1. Neural Network-Based Auditory for Audio Conversion

In general terms, automatic music transcription systems take an audio waveform as input, typically derived using its magnitude spectrogram $Z\in R^{+}$ (i.e., the short-time Fourier transform magnitude) and produce a pitch representation over time (termed *piano-roll algorithm* [48]. Firstly, the non-negative time–frequency representation of the input $Z\in R^{T\times F}$ is split into a product of two matrices: a dictionary $D\in R^{F\times \varrho }$ and an activation matrix $A\in R^{\varrho \times T}$, minimizing the distance d(·,·)∈R between Z and Y=DA. The optimizing framework for finding the non-negative matrices, D and A, can be expressed as follows:(1a)$$\begin{array}{cc} Y & =DA\approx Z \end{array}$$(1b)$$\begin{array}{cc} \; & s.t.:\min d(Y,Z)\to 0 \end{array}$$
where T∈N is the number of time points, F∈N is the number of spectral components, and ϱ∈N is the decomposition rank of Z.

Non-negative matrix factorization (NMF) provides a simple and effective approach to solving the minimization problem in Equations ([Disp-formula FD1a-sensors-25-01471]) and ([Disp-formula FD1b-sensors-25-01471]) using multiplicative update rules. In this framework, the spectrum at time point t∈T (the *t*-th row of Z) is modeled as a linear combination of ϱ spectral templates (columns of D). Consequently, a single spectral template represents the expected spectral energy distribution for each note produced by an instrument, while the corresponding activation row in A captures the intensity of that note’s playback over time [49].

#### 2.1.2. Autoencoder-Based Extraction of Piano-Roll Features

Next, the latent auditory feature set is extracted by applying semantic segmentation to *N* samples of the piano-roll representation obtained in the previous audio conversion step, expressed as $\left\{Y_{n}\in R^{h\times g\times c}|n\in N\right\}$, where *h* stands for height, *g* is width, and *c* is the number of image color channels (for simplicity, C1 is assumed).

A label mask, $M_{n}$, is generated from the estimated 2D image-like set, $Y_{n}$, to determine the class membership of each pixel in the *n*-th image. Each binary matrix mask, $M\in {0,1}^{h\times g}$, provides a two-label representation, indicating either the presence (i.e., 1) or absence (0) of a specific note in the piano-roll array. The label set is derived using autoencoder models enhanced with convolutional layers for semantic segmentation, as described in [50,51]:(2)$$\begin{array}{cc} \hat{M} & =(\varphi_{L}\left(\right)\circ \cdots \circ \varphi_{l}\left(\mu_{l-1}\right)\circ \cdots \circ \varphi_{1}\left(\right))\left\{Y\right\}\\ \; & \;\varphi_{l}:R^{h_{l-1}\times g_{l-1}\times D_{l-1}}\to R^{h_{l}\times g_{l}\times D_{l}},\;l\in L \end{array}$$
where the function composition for the *l*-th layer is $\varphi_{l}\left\{\cdot \right\}$, and $\mu_{l}=\varphi_{l}\left(\mu_{l-1}\right).$ is a single feature map at layer *l* that the neural network model produces in the manner outlined below:(3)$$\begin{array}{cc} \mu_{l} & =\psi_{l}\left(W_{l}⊛\mu_{l-1}+b_{l}\right)\in R^{h_{l}\times g_{l}\times D_{l}} \end{array}$$
where $W_{l}\in R^{\kappa_{l}\times \kappa_{l}\times D_{l-1}\times D_{l}}$ and $b_{l}\in R^{D_{l}}$ denote the learnable parameters that define the nonlinear activation function $\psi_{l}\left(\cdot \right)$. The parameter $\kappa_{l}$ specifies the dimensions of the *l*-th convolutional kernel, while $D_{l}$ represents the size of the extracted feature set. The symbol ⊛ denotes image-based convolution.

In turn, the following optimizing procedure estimates the parameter set $\Theta =\{W_{l},b_{l}\}$ needed in Equation (Equation 3) [52]:(4)$$\begin{array}{cc} \Theta^{*} & =\arg\min_{\Theta }E\left\{L\{M_{n},{\hat{M}}_{n}|\Theta \}:\forall n\in N\right\},\\ \; & \;L:{\left\{0,1\right\}}^{h\times g}\times {\left[0,1\right]}^{h\times g}\to R \end{array}$$
where L is the selected loss function, and E· is the notation for expectation operator.

Thus, to capture the most salient information at the *l*-th network layer, the function composition in Equation (Equation 2) transforms the input feature map from the preceding layer (l−1) into the output feature map $\mu_{l}$.

### 2.2. Supervised Extraction of Neural Responses

#### EEGNet-Based Feature Extraction

We assume that the emotion-prediction training space consists of two sets, X, Λ, where X contains EEG recordings, each spanning T∈N time instants and captured using a *C*-channel montage. Specifically, $X=X_{r}\in R^{C\times T}:r\in R$. Note that this procedure is performed in parallel with supervised auditory feature extraction, but employing the label set $\Lambda =\lambda_{r}\in {\left[0,1\right]}^{K}$ that encodes the emotional information with R∈N single-trial signals, classified according to an emotion recognition paradigm with a fixed number of emotional states, K∈N.

In this approach, a stack of spatial and temporal layers is employed to build the deep learning (DL) architecture for motor imagery (MI) classification, which predicts one-hot encoded class memberships by processing the input EEG data:(5)$$\hat{\lambda }=M\left(\xi_{L}\circ \cdots \circ \xi_{1}\right)\left\{X\right\},$$
where $M:R^{C\times T}\to {\left[0,1\right]}^{K}$ represents the mapping function (i.e., the neural network) applied to X, which performs a layer feature map with $P_{l}\in N$ elements at the *l*-th layer.

The same mapping process is defined as follows:$$\tilde{X_{l}}=\xi_{l}\left({\tilde{X}}_{l-1}⊗W_{l}+b_{l}\right),\;\tilde{X_{l}}\in R^{P_{l}}$$
where $\xi_{l}:R^{P_{l-1}}\to R^{P_{l}}$ denotes a learning function incorporating a nonlinear activation, $W_{l}\in R^{P_{l-1}\times P_{l}}:l\in L$ represents the set of layer weights, $b_{l}\in R^{P_{l}}$ is the bias term, and L∈N indicates the network depth. The symbols ∘ and ⊗ denote function composition and a given tensor operation, respectively, such as a fully connected product or 1D/2D convolution.

Likewise, the estimation framework described in Equation (Equation 5) relies on optimizing the parameter set $\Theta =W_{l},b_{l}$, analogous to the earlier case of auditory feature extraction. Consequently, the following optimization process is performed across the trial set:(6)$$\Theta^{*}=\arg\min_{\Theta }E\left\{L\left(\lambda_{r},{\hat{\lambda }}_{r}|\Theta \right)+\gamma \Omega \left(\Theta \right):\forall r\in R\right\},$$
where the loss function $L:{0,1}^{K}\times {\left[0,1\right]}^{K}\to R$ measures the prediction error, the regularization function Ω(·) introduces constraints to avoid overfitting, and the trade-off term $\gamma \in R^{+}$ balances fitting the data against incorporating prior knowledge. Notably, we define ${\tilde{X}}_{0}=X$, ${\tilde{X}}_{L}=\hat{\lambda }$, and employ a specific sigmoid function as the nonlinear activation to facilitate supervised feature extraction.

To enhance the classification of neural responses, the optimization process outlined in Equation (Equation 6) is executed within the EEG-Net framework. This framework functions as a compact convolutional network, enabling convolutional kernel connectivity between input and output feature maps.

Therefore, the EEGNet pipeline begins with temporal convolution to construct a set of frequency filters, as detailed in [53]. This is followed by depth-wise convolution applied to each feature map to learn spectro-spatial filters. Subsequently, a separate convolution captures temporal summaries, and a point-wise convolution determines how to combine the feature maps for class membership prediction.

### 2.3. Affective-Based Prediction of MIDI Data

At this stage, the extracted auditory features $\left\{Y_{n}\right\}$ are linked with computed EEG patterns in a supervised manner using the affective label set Λ. To evaluate the relationship between the extracted auditory features and the EEG patterns, the outputs of the deep model are processed through the Centered Kernel Alignment algorithm. However, given the significant variability in the extracted features—primarily the high variation in predicted emotion labels—we adapt a convolutional neural network, initially developed for BCI classification tasks, into a regression framework commonly employed for emotion recognition.

#### 2.3.1. Kernel-Based Alignment Between Supervised Auditory and Neural Features

Using the available affective label set, Λ, we identify closer associative similarities. Specifically, we apply Centered Kernel Alignment (CKA) to the two extracted feature sets: the autoencoder-based piano-roll features, $\tilde{Y}\in Y$, and the EEGNet-based features, $\tilde{X}\in X$, obtained through supervised extraction. These features are extracted in a supervised manner, utilizing the available affective label set, Λ. To achieve this, the nonlinear relationships among samples in each space are encoded using corresponding positive-definite matrices (kernels): $K_{\tilde{X}}:X\times X$ and $K_{\tilde{Y}}:Y\times Y$. As detailed in [54] regarding the sonification of affective neural responses, both feature matrices are computed within the framework of reproducing kernel Hilbert spaces.

As further detailed in [55], the empirical Centered Kernel Alignment that is denoted as ${\hat{\rho }}_{CKA}\left(\cdot ,\cdot \right)\in R\left[0,1\right]$ measures the similarity between the two random sets $\tilde{X}$ and $\tilde{Y}$. This real-valued alignment is estimated as follows:$$\begin{array}{cc} {\hat{\rho }}_{CKA}\left(K_{\tilde{X}},K_{\tilde{Y}}\right) & =\frac{{\langle {\bar{K}}_{\tilde{X}},{\bar{K}}_{\tilde{Y}}\rangle }_{F}}{\sqrt{∥{\bar{K}}_{\tilde{X}}{∥}_{F}{∥{\bar{K}}_{\tilde{Y}}∥}_{F}}}, \end{array}$$
where the centered kernel matrix is computed as ${\bar{K}}_{Z}=H{\bar{K}}_{Z}H^{⊤}$ with $H=I-\frac{1}{N_{h}}1_{N}h1_{N}h^{⊤}$, $I_{N}h$ is the identity matrix, and $1_{N}h$ as the all-one vector the size of $N_{h}$ ($N_{h}$ is the cardinal of trace{H}). Notations ${∥\cdot ∥}_{F}$ and ${\langle \cdot ,\cdot \rangle }_{F}$ stand for the Frobenius norm and inner product, respectively.

#### 2.3.2. *k*-Neighbors-Based Emotion Label Prediction of MIDI Representations

Given the emotional complexity of music, particularly in intricate examples, and the continuous nature of emotions, recognition models that provide a more detailed depiction of subjective experiences are favored over those that reduce affective states to a limited set of categories [56].

It is important to note that models, due to their high nonlinearity, are inherently black-box and lack interpretability. To address this, the introduction of the *k*-nearest neighbors (*k*-NN) algorithm as a direct interpretability procedure enables better alignment of the matched spaces. In this context, we utilize the (*k*-NN) algorithm to identify potential emotional neural responses elicited by music clips and predict their closest MIDI representations, leveraging the simplicity and generalizability of the prediction models involved.

Specifically, the latent space extracted by the encoder is compared with the clustered EEGNet features. Using $\lambda_{r}\in \Lambda$ as the original labels for arousal and valence, and $\hat{\lambda }$ as the EEGNet predictions, the Euclidean distance is calculated as $\lambda_{\Delta }={\left|\lambda_{r}-\hat{\lambda }\right|}_{2}$. This distance is then processed through a softmax activation function:(7)$$\mathrm{softmax}\left(\lambda_{\Delta }\right)=\frac{e^{\lambda_{\Delta }}}{\sum_{i=1}^{n}e^{\lambda_{\Delta_{i}}}}$$

Afterwards, we define $X_{code}$ as:(8)$$X_{code}=\mathrm{softmax}\left(\lambda_{dist}\right)\cdot Y_{code}$$
where $Y_{code}$ is the output of the piano-roll encoder.

It should be noted that the clustering technique described above is applied to each examined individual.

## 3. Experimental Set-Up

### 3.1. Evaluating Framework and Tested Dataset

In this work, we present a deep learning approach for EEG-based emotion prediction triggered by music clips, incorporating supervised feature extraction for MIDI generation Specifically, the proposed method is evaluated using the DEAP Dataset for Emotion Analysis. The evaluation process for generating auditory data from affective EEG representations comprises the following stages (Figure 1):(i)The segment-wise preprocessing and time-windowing of the training data: This stage involves dividing EEG signals into non-overlapping or overlapping time segments to ensure a structured representation of temporal features. The preprocessing steps include filtering, artifact removal, normalization, and noise reduction to enhance signal quality. The resulting time-windowed data serve as input for subsequent feature extraction.(ii)Supervised deep feature extraction for emotion classification: This step involves computing EEG neural responses using the EEGNet framework, a convolutional neural network specifically designed for EEG signal analysis. The extracted features are then used to classify different emotional states. Additionally, auditory features are derived from music clips using the piano-roll algorithm, which structures music data into the pitch, velocity, and timing characteristics of musical notes.(iii)Affective-based MIDI prediction and feature alignment: In this stage, EEG-derived features are aligned with auditory features from music clips to generate MIDI representations that reflect the participant’s emotional responses. Machine learning techniques are employed to correlate EEG signals with emotional cues embedded in the auditory data. The final output is a MIDI prediction representing affective states derived from EEG patterns.

#### 3.1.1. EEG Data Description

This evaluated database is publicly available at https://www.eecs.qmul.ac.uk/mmv/datasets/deap/ (accessed on 1 November 2024) and contains EEG recordings and peripheral physiological signals collected from *M* = 32 participants who viewed 40 one-minute music video excerpts. After watching each excerpt, each subject rated the video according to the four emotion conditions: arousal, valence, like/dislike, dominance, and familiarity. As described in [57], the EEG paradigm is based on stimulus selection through affective tag retrieval, video highlight detection, and an online evaluation tool.

From each subject, the EEG signal at 512 Hz cut-off frequency was recorded in a 32-channel montage consisting of the following electrodes: Fp1, AF3, F3, F7, FC5, FC1, C3, T7, CP5, CP1, P3, P7, PO3, O1, Oz, Pz, Fp2, AF4, Fz, F4, F8, FC6, FC2, Cz, C4, T8, CP6, CP2, P4, P8, PO4, and O2.

In the preprocessing step, the data were downsampled to 128 Hz, and artifacts, including EOG signals, were removed as performed in [57]. A band-pass filter was then applied to retain frequencies in the 4–45 Hz range. Common-reference spatial filtering was performed to mitigate volume-conduction effects. The filtered signals were segmented into 60 s trials, excluding a 3-s pre-trial baseline. Finally, the EEG channels were reordered according to the Geneva Order for easier interpretation.

#### 3.1.2. Auditory Data Description

This dataset includes a collection of music-video clips curated to elicit specific mental states associated with the emotion space [58]. The acquisition protocol began with a 2-min baseline recording in a relaxed state. During each trial, participants listened to a one-minute audio recording. Two sessions, each consisting of 20 trials, were conducted, separated by a break to verify the quality of the acquired data and ensure proper electrode placement. At the end of each trial, participants rated their emotional levels, categorized as arousal, valence, liking, and dominance.

For evaluation, each one-minute trial was divided into ten non-overlapping segments of music and EEG data, each lasting six seconds. To capture more musical information, this interval was set longer than the four-second duration used in [59]. Additionally, nine audio segments with fade-out endings were excluded, resulting in 391 valid testing segments instead of the original 400.

Note that each of the 391 validation segments represents a six-second EEG recording aligned with its corresponding MIDI segment. The dataset was split into an 80% training set and a 20% testing set for cross-validation.

### 3.2. Enhanced Feature Extraction Using Deep Learning Models

#### 3.2.1. Deep Feature Extraction from EEG Data

In this study, we adopt the EEGNet framework, a widely used deep learning model for EEG-based emotion recognition, specifically for predicting arousal and valence. In this paradigm, the objective is to predict continuous values within the arousal–valence space, a commonly used representation in affective computing. Arousal refers to the intensity of an emotional response, ranging from calm to excited, while valence reflects emotional polarity, from negative to positive. This approach provides a more fine-grained representation of affective states, as opposed to relying on discrete categorical classifications.

To address significant inter-segment variability, dimensionality reduction is applied to the extracted feature sets to identify the optimal temporal segments for predicting affective labels. Table 1 outlines the configuration of the EEGNet framework used in this study. Notably, the outputs (highlighted layers) are specifically configured to enable the neural network to function as a regression model.

#### 3.2.2. Deep Feature Extraction from MIDI Data

Initially, the 40 one-minute audio clips from the DEAP dataset are transcribed into MIDI format using an automatic music transcription method with a multi-output structure to enhance frame-level note accuracy, as described in [60]. The transcription is performed using the default tool parameters: note segmentation is set to 0.5 s, the model confidence threshold to 0.3, the minimum note length to 11, and the MIDI tempo to 120 beats per minute.

Next, convolutional deep learning models are applied, requiring 2D representations of music scores. These are created by converting each score over time segments into an evolving image array, known as the pitch or *piano-roll* representation. In this work, the pretty-midi package is used to generate a binary-valued mask by thresholding the incoming audio data, as described in [61]. Subsequently, the required MIDI segmentation for feature extraction is performed on the generated binary mask using a variant of the U-Net neural network, referred to as the *AutoEncoder Piano Roll*.

Table 2 presents the fixed configuration of the U-Net architecture, where the residual layers (highlighted in bold) are connected to the add layer instead of being arranged sequentially. The implementation code for the AutoEncoder Piano Roll is publicly accessible at https://github.com/hdperezn/Autoencoder_Piano_Roll (accessed on 1 November 2024).

### 3.3. Proposed Deep Alignment Between Enhanced EEG and MIDI Feature Sets

At this stage of data alignment, a regression model is employed to align the extracted feature sets from EEGNet and the autoencoder. However, the following constraints are imposed when working with the estimated feature sets:(a)The label scores produced by the aforementioned feature-extraction algorithms are binary, making them insufficient to capture the heterogeneity between the spaces being aligned.(b)The derived MIDI sets tend to present high variability when representing each emotion.

To address these limitations, we propose aligning the spaces through a linear combination of the enhanced feature sets using *k*-nearest neighbor representations. Specifically, the *k* nearest embedded EEG features are identified for emotion score prediction by extracting the corresponding MIDI feature sets, following these steps:The deep learning framework extracts the most accurate segment-wise feature sets for predicting affective scores. To function as a real-valued score predictor, the softmax output layer is divided into two separate output layers: one for predicting valence ($\delta_{valence}$) and another for arousal ($\delta_{arousal}$). Each layer uses a sigmoid activation function and is trained separately using Mean Squared Error (MSE) loss.Similarly, for EEG data, the feature-extraction process is enhanced by leveraging the subjects’ labels to obtain piano-roll representations of the musical stimuli that best distinguish the affective tags. To achieve this, the selected autoencoder architecture is operated in a supervised mode by incorporating Centered Kernel Alignment (CKA) as described in Equation [eq]. This approach forces the bottleneck layer to learn the most discriminative feature sets. In essence, the piano-roll algorithm reconstructs the MIDI representation, extracting the feature sets that provide the greatest separation between emotional states.

Therefore, the estimated neighbors are mapped onto the autoencoder space, enabling the identification of the autoencoder’s embedded representation corresponding to each EEG signal. The algorithm for the proposed enhanced alignment is outlined in Algorithm 1.
**Algorithm 1** The prediction of labels generated by the piano-roll features.  1:**procedure** GeneratePianoRoll($y_{\mathrm{pred}\_\mathrm{valence}},y_{\mathrm{pred}\_\mathrm{arousal}},y_{\mathrm{continuos}\_\mathrm{valence}},y_{\mathrm{continuos}\_\mathrm{arousal}},train\_index,cka\_pr,X_{\mathrm{pr}\_\mathrm{train}}$)  2:      **Initialization:**  3:      $y_{\mathrm{pred}\_\mathrm{array}}\leftarrow \mathrm{Concatenate}\left(y_{\mathrm{pred}\_\mathrm{valence}},y_{\mathrm{pred}\_\mathrm{arousal}},\mathrm{axis}=1\right)$  4:      $y_{\mathrm{true}\_\mathrm{array}}\leftarrow \mathrm{Concatenate}\left(y_{\mathrm{continuos}\_\mathrm{valence}}\left[train\_index\right],y_{\mathrm{continuos}\_\mathrm{arousal}}\left[train\_index\right],\mathrm{axis}=1\right)$  5:      **Distance Calculation:**  6:      $y_{\mathrm{dist}}\leftarrow \mathrm{CalculateDistances}\left(y_{\mathrm{pred}\_\mathrm{array}},y_{\mathrm{true}\_\mathrm{array}},‘\mathrm{euclidean}’\right)$  7:      $dense\_ydist\leftarrow \mathrm{SOFTMAX}(y_{\mathrm{dist}})$  8:      **Index Selection:**  9:    max_indices←Top5Indices(−dense_ydist.numpy(),axis=1)10:      result_array←CreateResultMatrix(dense_ydist.numpy(),max_indices)11:      **EEG Encoding:**12:      $X_{\mathrm{pr}\_\mathrm{code}}\leftarrow \mathrm{Encode}\left(X_{\mathrm{pr}\_\mathrm{train}}\left[train\_index\right]\right)$13:      $eeg\_coding\leftarrow \mathrm{MatrixMultiply}(result\_array,X_{\mathrm{pr}\_\mathrm{code}})$14:      **Decoding:**15:      $X_{\mathrm{pr}\_\mathrm{pred}}\leftarrow \mathrm{Decode}\left(eeg\_coding,cka\_pr\right)$16:      **Return:** $X_{\mathrm{pr}\_\mathrm{pred}}$17:**end procedure**

Figure 2 illustrates the impact of the CKA loss on structuring the latent space of the MIDI autoencoder within the emotion recognition framework. Figure 2a displays the true emotion labels for each stimulus, where each dot represents the arousal (*x*-axis) and valence (*y*-axis) ratings assigned by Subject #1, highlighting the continuous nature of emotion classification in the arousal–valence space. Figure 2b shows the MIDI latent space when the autoencoder is trained solely with a reconstruction loss, visualized using a t-SNE projection. Here, the representations are organized for MIDI reconstruction but lack a clear emotional structure. In contrast, Figure 2c demonstrates how incorporating CKA loss introduces a more distinct organization of the latent space according to arousal and valence. This alignment, achieved by enforcing similarity with the EEG domain, not only preserves reconstruction performance but also enhances interpretability by maintaining an emotion-based structure. Overall, Figure 2 highlights the effectiveness of CKA loss in aligning the MIDI embedding space with emotional dimensions, thereby enabling more emotion-driven music generation.

### 3.4. Evaluation Performance

As outlined in [21], both musical and neurophysiological dimensions are integrated to comprehensively evaluate the proposed system using the following metrics:

#### 3.4.1. Musical Evaluation Metrics

These metrics, derived from the MIDI feature-extraction process, assess the quality and structure of the generated musical compositions:–*Pitch Class (PC):* This metric evaluates the distribution of note pitches throughout a musical composition, based on the concept of pitch classes, where notes are grouped according to their relation to the musical scale. The PC metric is significant as it reveals modulation and tonality in the generated music. Analyzing the PC can help identify melodic and harmonic patterns that align with the intended musical objectives in MIDI generation.–*Pitch Range (PR)*: The pitch range reflects the amplitude span of notes used in a composition, serving as a measure of tonal diversity and richness. A broader range indicates greater note variability, enhancing musical texture and auditory interest. When linked to EEG signals, PR provides insights into the composer’s emotional and cognitive variability during the creative process.–*Inter-Onset Interval (IOI)*: This metric examines the time between the onset of one note and the start of the next, offering insights into the rhythmic structure of a composition. The IOI defines the flow and cadence of music. Its detailed analysis can uncover rhythmic patterns and dynamics essential for musical interpretation. In the context of EEG signals, IOI reflects the composer’s attention and focus during music creation, influencing the resulting compositions.–*Pitch-Class Histogram (PCH)*: The PCH visually represents the frequency of occurrence for each pitch class in a composition. This metric highlights trends in pitch usage, identifying dominant notes and their relationships. Analyzing the PCH can uncover the harmonic and melodic structure of the generated music, offering valuable insights into the composer’s musical preferences and decisions.

#### 3.4.2. Performance Classification Measures of Neural Responses

For EEGNet-based feature extraction, the model’s performance in classifying EEG data is evaluated using three key metrics:–*Accuracy*: This metric calculates the proportion of correctly classified instances out of the total number of samples, serving as a baseline indicator of the model’s predictive performance.–*Cohen’s Kappa (κ)*: This statistic measures the agreement between predicted and actual labels while accounting for chance agreement, providing a more robust evaluation compared to accuracy alone.–*Area Under the Curve (AUC)*: This metric assesses the model’s ability to differentiate between classes by summarizing its performance across all classification thresholds. Together, these metrics offer a comprehensive evaluation of the EEGNet model’s effectiveness in feature extraction and classification.

## 4. Results and Discussion

In this paper, we present three experiments to demonstrate the model’s capability to classify emotion categories and reconstruct MIDI sequences by analyzing musical characteristics and EEG data.

First, we assess the EEGNet-based model’s ability to classify EEG signals into arousal and valence categories, validating its effectiveness for such tasks. Performance metrics are provided, showing results that are competitive with state-of-the-art models.

Second, we evaluate the system’s ability to reconstruct musically coherent MIDI sequences. This involves comparing the original and reconstructed sequences, with results highlighting the model’s effectiveness in the MIDI reconstruction task.

Lastly, we investigate subject-specific influences on output quality using metrics such as KL divergence and overlap area. Collectively, these experiments highlight the robustness of the proposed approach, offering a comprehensive evaluation of its ability to classify EEG signals and generate musically coherent MIDI content.

### 4.1. Experiment 1: Comparison with State-of-the-Art Models

To evaluate the performance of the EEGNet model used in this study, a comparative analysis was conducted against FusionNet, a state-of-the-art model commonly employed in EEG classification tasks. FusionNet was selected for its prominence in the field and its reported effectiveness in similar scenarios. Table 3 presents the evaluation metrics, including accuracy and Cohen’s kappa, chosen to assess both the predictive performance of the models and the agreement between predicted and true labels.

The results demonstrate that EEGNet achieved competitive performance, with an average accuracy of 75.0±14.1%, compared to 57.7±15.3% for FusionNet. This indicates that EEGNet effectively classifies EEG signals with high precision. Furthermore, EEGNet achieved a Cohen’s kappa score of 50.9±28.3%, compared to 45.7±19.8% for FusionNet, reflecting a strong agreement between the model’s predictions and the ground truth, thereby confirming its reliability.

While previous claims of robustness for FusionNet were based on its performance across various datasets, this study focused exclusively on a single dataset to ensure a controlled and consistent evaluation framework. Although these results highlight EEGNet’s potential, further testing on diverse datasets is necessary to confirm its robustness and generalizability.

These findings establish EEGNet as a competitive model for EEG signal classification in controlled experimental conditions, offering strong predictive capabilities and reliable agreement with true labels.

Overall, while more complex models, such as large-scale transformer-based architectures or diffusion models, could enhance feature extraction and improve MIDI reconstruction, their effectiveness heavily relies on the availability of high-quality, large-scale EEG datasets with reliable annotations. The inherent noise, inconsistencies in labeling, and subject-dependent variability in existing EEG datasets make it challenging to justify the implementation of such resource-intensive techniques [63,64].

### 4.2. Experiment 2: MIDI Reconstruction

We also assess the system’s capability to reconstruct musical stimuli using a diverse set of MIDI inputs. This evaluation focuses on how well the reconstructed outputs preserve critical musical attributes such as pitch, rhythm, harmony, and overall structural integrity. Figure 3 presents the results of this analysis, offering a visual comparison between the original MIDI sequences (top panel) and their corresponding reconstructions (bottom panel). The clear alignment between the sequences highlights the model’s effectiveness in maintaining key musical features during the reconstruction process.

Quantitative evaluation was performed using the Dice coefficient, which measures the similarity between the original and reconstructed sequences. The analysis yielded an average Dice score that tends to small values, indicating a high level of accuracy in replicating the original MIDI structures. This metric highlights the model’s ability to maintain both temporal alignment and tonal consistency, even when processing complex or dense musical inputs. Additionally, the system exhibited minimal reconstruction errors, further demonstrating its reliability in handling intricate musical patterns.

Notably, the model effectively reconstructed complex MIDI inputs in challenging scenarios, such as rapid note transitions, overlapping melodies, or dynamic variations—areas that often pose significant challenges for generative models. The system’s resilience in these situations underscores its versatility and adaptability across diverse musical contexts, making it well suited for a wide range of applications [65].

These findings are pivotal, as they establish a solid foundation for leveraging the system in subsequent phases of the study. By demonstrating its proficiency in preserving critical attributes of musical stimuli during reconstruction, the model has proven to be a reliable tool for generative tasks [66]. This capability is essential for the next phase, where the reconstructed MIDI sequences will be utilized for real-time music generation and adaptive music systems.

Thus, the results validate the proposed model’s effectiveness in handling MIDI data with precision and robustness. The demonstrated reconstruction accuracy paves the way for its integration into advanced applications, bridging the gap between computational techniques and human-perceived musical quality [67].

### 4.3. Experiment 3: Performance Evaluation Across Subjects

This experiment aimed to evaluate how individual subject variability influences the quality and characteristics of the generated outputs. The analysis investigated the impact of inter-subject differences in EEG signals on the system’s ability to produce consistent and high-quality MIDI sequences. Both visual and quantitative assessments were performed.

Figure 4 and Figure 5 provide graphical representations of the variability observed in the generated outputs. Figure 4 displays the probability density functions (PDFs) of key acoustic metrics for each subject, illustrating the range and distribution of outputs. Figure 5 complements this by offering a compact visualization of metric distributions, emphasizing differences in central tendency and variability across subjects. Together, these visualizations highlight patterns of individual subject variability and their influence on the model’s generative performance.

Additionally, a detailed comparison of acoustic metrics was conducted across three distinct scenarios:–*Training versus Training (T vs. T)*: Evaluates the consistency of outputs generated from EEG data within the training group, serving as a baseline.–*Training versus Worst-Performing Subject (T vs. W)*: Assesses the impact of the worst-performing subject’s EEG data on the quality of generated MIDI sequences.–*Training versus Best-Performing Subject (T vs. B)*: Highlights the influence of the best-performing subject’s EEG data, offering insights into the system’s upper performance limits.

For each scenario, the results of these comparisons are presented in Table 4, respectively. For each comparison, metrics such as mean and standard deviation (to summarize central tendencies and variability), KL divergence (to quantify differences between feature distributions), and overlap area (to assess distribution similarity) were calculated.

The findings reveal significant variability in how individual subjects influence the quality of generated MIDI sequences:–In the *T vs. T group*, metrics demonstrate high consistency, reflecting the system’s ability to generalize across training data.–In the *T vs. W comparison*, increased KL divergence and reduced overlap area indicate that EEG variability from the worst-performing subject introduces challenges in maintaining output quality.–Conversely, *the T vs. B analysis* shows reduced divergence and higher overlap, suggesting that EEG signals from the best-performing subject align closely with the training data, enabling the system to generate more coherent and high-quality outputs [68].

These results underscore the critical role of subject-specific EEG variability in shaping the performance of EEG-based generative systems [68]. The robust evaluation framework applied in this experiment provides a comprehensive understanding of how inter-subject differences affect system outputs, paving the way for refinements in training protocols and model optimization strategies.

## 5. Concluding Remarks

This study presented a novel framework for generating musically coherent MIDI sequences from EEG-based emotion predictions, utilizing deep learning architectures to bridge neural and auditory domains. The approach integrates advanced feature-extraction techniques, including EEGNet for emotion classification and an autoencoder-based piano-roll representation for auditory feature generation. These components were aligned using Centered Kernel Alignment (CKA) to establish robust mappings between EEG-derived features and their corresponding musical outputs.

Nonetheless, we would like to highlight the following key aspects of our proposal:*Emotion Classification Performance:* The EEGNet model demonstrated strong performance in classifying emotional states (arousal and valence) from EEG signals, achieving an average accuracy of 75.0±14.1% and a Cohen’s kappa score of 50.9±28.3%. These results surpass state-of-the-art models such as FusionNet, highlighting the model’s reliability in distinguishing emotional states with precision. The focus on a single dataset ensured consistent performance in controlled experimental conditions, setting a robust benchmark for emotion classification.*MIDI Reconstruction Evaluation*: The system’s ability to reconstruct MIDI sequences was rigorously assessed using musical metrics such as pitch class (PC), pitch range (PR), inter-onset interval (IOI), and pitch-class histogram (PCH). The results showed that the reconstructed sequences preserved critical musical features, maintaining temporal alignment and tonal consistency. The system exhibited resilience in handling challenging scenarios, such as rapid note transitions, overlapping melodies, and dynamic variations. High Dice similarity scores between the original and reconstructed MIDI sequences confirmed the system’s effectiveness in retaining musical attributes and structural integrity, making it suitable for generative and adaptive music applications.*Subject-Specific Variability*: A detailed analysis of subject-specific variability revealed how individual differences in EEG signals influence output quality. Metrics such as KL divergence and overlap area indicated that worst-performing subjects introduced greater divergence and reduced overlap, challenging output consistency. Conversely, the best-performing subjects exhibited closer alignment with training data, producing more coherent and higher-quality MIDI outputs. These findings highlight the importance of adaptive training protocols to account for inter-subject differences, enhancing generalizability and consistency.*CKA-Based Alignment*: The CKA-based alignment technique proved effective in extracting highly discriminative features from both neural and auditory spaces. This alignment improved the separation of affective labels and enhanced the quality of the reconstructed MIDI sequences. Visual tools such as probability density functions (PDFs) and violin plots illustrated the system’s adaptability and performance across diverse EEG profiles, reinforcing its robustness.

Although the proposed framework is validated on real-world data, its applicability to real-world music generation still requires further refinement. To develop a methodology competitive with state-of-the-art generative models that do not rely on EEG inputs, the first step is to establish a larger and more robust dataset. Such a dataset would enable integration with more advanced generative architectures, such as diffusion-based models or autoregressive transformers, which require extensive training data to produce high-quality musical outputs. However, the current dataset size and inter-subject variability impose limitations on directly implementing these models, making our approach an effective yet computationally feasible alternative under the given conditions.

Another factor limiting its real-world application is the high intra-subject variability, particularly among individuals with limited proficiency in BCI usage. This underscores the need for adaptive training protocols to enhance generalizability and optimize system performance.

In conclusion, and elucidating future directions, this research represents a significant advancement in integrating brain–computer interface technologies with generative music systems. By achieving competitive classification accuracy, robust MIDI reconstruction, and adaptability to individual EEG variability, the proposed framework lays a strong foundation for future innovations.

Potential directions for further exploration include: (i) Expanding the dataset diversity to improve generalizability; (ii) Optimizing neural architectures for enhanced scalability and performance; and (iii) Exploring real-time applications in adaptive and interactive music generation systems.

These efforts will continue to bridge computational techniques with human emotional expression, advancing the fields of affective computing and human–computer interaction while enriching the musical experience.

Based on the study’s findings and identified limitations, future work can enhance the framework by focusing on:–*Data Diversity and Model Refinement*: Expanding EEG datasets with diverse demographics, emotions, and experimental conditions (e.g., varied stimuli and EEG setups) is crucial for robustness. Furthermore, refining deep learning architectures with lighter, scalable networks—enhanced with regularization or self-supervised learning—can enable real-time implementations in resource-constrained environments.As stated in [9], our DL proposal for music generation can be refined in terms of both neurophysiological models for emotion perception and musical structures, highlighting key areas for further cognitive and neurophysiological research.–*User Customization and Musical Exploration*: Tailoring the system with adaptive training and hybrid models to accommodate individual EEG variability will improve accuracy and musical coherence. Exploring diverse musical formats beyond piano roll (e.g., varied instrumentation, complex rhythms) can broaden the model’s capabilities.–*Real-time Applications and User Evaluation*: Developing live BCI-based interactive music systems that adapt to real-time EEG input will expand use cases. The integration of multi-sensor data (e.g., heart rate) could enrich affective inferences. Extensive user evaluations and psychological studies are needed to validate the perceived emotional quality of generated music, including its potential for therapy and mood regulation.

## Figures and Tables

**Figure 1 sensors-25-01471-f001:**
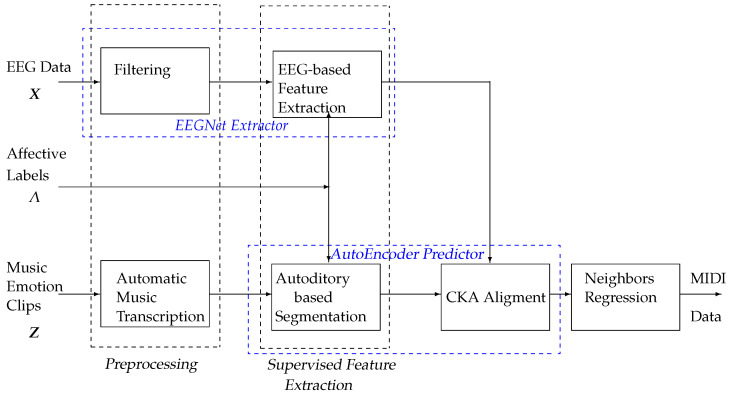
Proposed deep learning framework for EEG-based emotion prediction using supervised feature extraction for MIDI generation. Stages: (i) segment-wise preprocessing; (ii) supervised deep feature extraction for emotion classification; and (iii) affective-based MIDI prediction and feature alignment.

**Figure 2 sensors-25-01471-f002:**
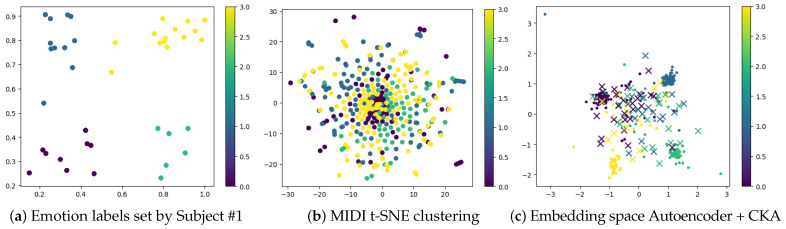
Visualization of emotion labels and MIDI feature representations. (**a**) Emotion labels set by Subject #1, where the *x*-axis represents arousal and the *y*-axis represents valence. (**b**) Two-dimensional t-SNE projection (n_components=2, perplexity =5) of the piano-roll arrays, illustrating clustering of MIDI features based on emotion labels. Colors indicate the class of each audio stimulus. (**c**) Embedding space obtained from the bottleneck representation of the piano-roll autoencoder trained with CKA loss. Dots correspond to training data, while crosses (×) represent test data. Of note, the axes are resized to provide better visual perception of the plotted values.

**Figure 3 sensors-25-01471-f003:**
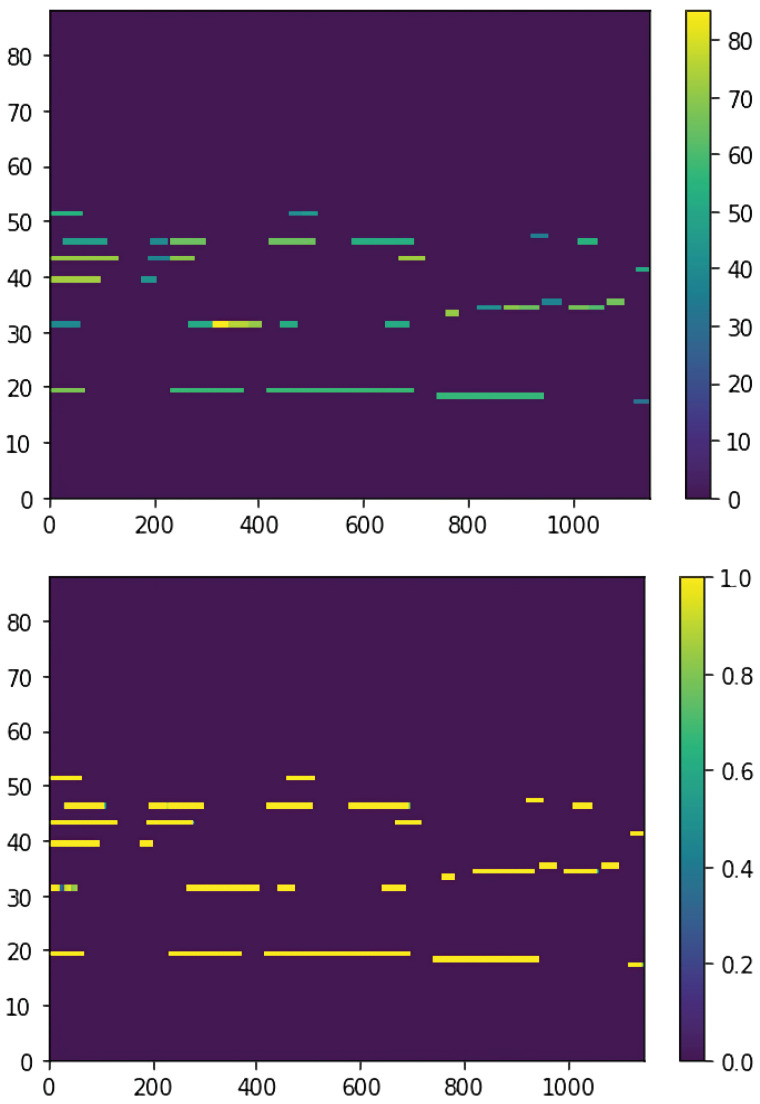
Comparison between original MIDI (**top**) and reconstructed MIDI (**bottom**). Of note, the axes are resized to provide better visual perception of the plotted values.

**Figure 4 sensors-25-01471-f004:**
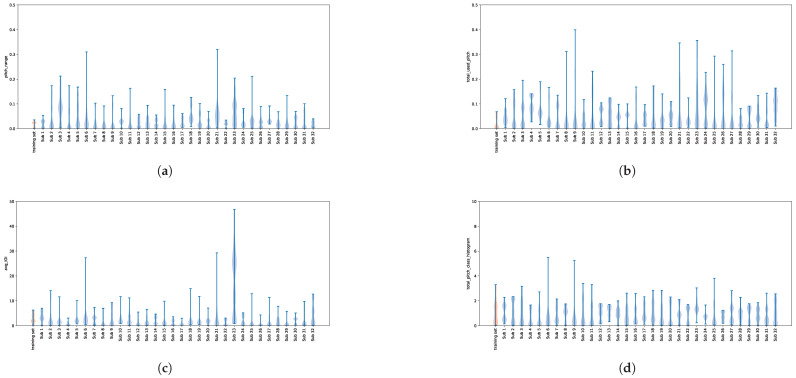
Probability density functions (PDFs) of model characteristics for all subjects in fold 1. Each subfigure corresponds to a specific feature extracted from the MIDI data. (**a**) Feature 0: description of the feature (e.g., pitch range). (**b**) Feature 1: description of the feature (e.g., total used pitch). (**c**) Feature 2: description of the feature (e.g., average IOI). (**d**) Feature 3: description of the feature (e.g., pitch-class histogram).

**Figure 5 sensors-25-01471-f005:**
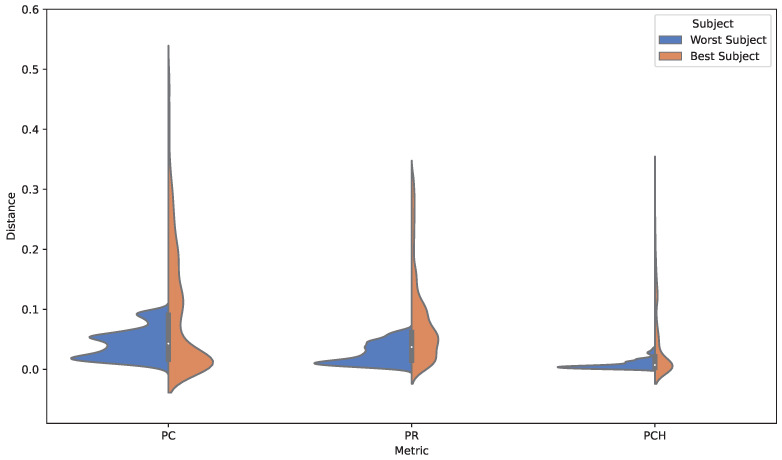
Violin plots comparing metrics for the best- and worst-performing subjects.

**Table 1 sensors-25-01471-t001:** Detailed EEGNet architecture for MI classification.

Layer	Output Dimension	Parameters
Input Layer	$N_{c}\times N_{t}\times 1$	·
Conv2D	$N_{c}\times N_{t}\times F_{1}$	Temporal filter ($F_{1}$) = 4, Kernel size = (1, 4) Padding = same, Bias = False
Batch Normalization	·	·
DepthwiseConv2D	$1\times N_{t}\times 16$	Depth Multiplier (*D*) = 2, Kernel size = ($N_{c}$, 1) Bias = False
Batch Normalization	·	·
Activation	·	Activation = ELU
AveragePooling2D	1×32×16	Pool size = (1, 4)
Dropout	·	Dropout Rate = 0.6
SeparableConv2D	1×32×32	Temporal filter ($F_{2}$) = 32, Kernel size = (1, 16) Padding = same, Bias = False
Batch Normalization	·	·
Activation	·	Activation = ELU
AveragePooling2D	1×4×32	Pool size = (1, 8)
Dropout	·	Dropout Rate = 0.6
Flatten	128	·
Dense	$N_{\mathrm{units}}$	Units = $N_{\mathrm{classes}}$ or 2
Activation	$N_{\mathrm{units}}$	Activation = Softmax or Sigmoid

Notations: $N_{c}$: number of EEG channels; $N_{t}$: number of time samples; $F_{1}$, $F_{2}$: number of filters; D: depth multiplier for depth-wise convolution; ELU: Exponential linear unit activation function; Dropout: proportion of neurons deactivated; Softmax/Sigmoid: activation functions for multi-class or binary classification.

**Table 2 sensors-25-01471-t002:** AutoEncoder Piano Roll—Encoder.

Layer	Output Dimension	Parameters
Input Layer	$N_{p}\times N_{t}\times 1$	·
Conv2D	$\frac{N_{p}}{2}\times \frac{N_{t}}{2}\times F_{1}$	$F_{1}=32$, Kernel size = 1 Padding = same, Strides = 2
BatchNormalization	·	·
Activation	·	Activation = ReLU
Residual (Conv2D)	$\frac{N_{p}}{4}\times \frac{N_{t}}{4}\times F_{2}$	$F_{2}=64$, Kernel size = 3 Padding = same, Strides = 2
Dropout	·	Dropout Rate = 0.5
Activation	·	Activation = ReLU
SeparableConv2D	$\frac{N_{p}}{2}\times \frac{N_{t}}{2}\times F_{2}$	$F_{2}=64$, Kernel size = 3 Padding = same, Bias = False
BatchNormalization	·	·
Activation	·	Activation = ReLU
SeparableConv2D	$\frac{N_{p}}{2}\times \frac{N_{t}}{2}\times F_{2}$	$F_{2}=64$, Kernel size = 3 Padding = same, Bias = False
BatchNormalization	·	·
MaxPooling2D	$\frac{N_{p}}{4}\times \frac{N_{t}}{4}\times F_{2}$	Pool size = 3, Strides = 2
Add [MaxPooling2D, Residual]	·	·
Residual (Conv2D)	$\frac{N_{p}}{8}\times \frac{N_{t}}{8}\times F_{3}$	$F_{3}=128$, Kernel size = 3 Padding = same, Strides = 2
Dropout	·	Dropout Rate = 0.5
Activation	·	Activation = ReLU
SeparableConv2D	$\frac{N_{p}}{4}\times \frac{N_{t}}{4}\times F_{3}$	$F_{3}=128$, Kernel size = 3 Padding = same, Bias = False
BatchNormalization	·	·
Activation	·	Activation = ReLU
SeparableConv2D	$\frac{N_{p}}{4}\times \frac{N_{t}}{4}\times F_{3}$	$F_{3}=128$, Kernel size = 3 Padding = same, Bias = False
BatchNormalization	·	·
MaxPooling2D	$\frac{N_{p}}{8}\times \frac{N_{t}}{8}\times F_{3}$	Pool size = 3, Strides = 2
Add [MaxPooling2D, Residual]	·	·

Notations: $N_{p}$: number of pitch classes; $N_{t}$: number of time steps; $F_{1}$, $F_{2}$, $F_{3}$: number of filters applied in each block; kernel size: size of the convolutional filters; strides: step size for the convolution; dropout rate: proportion of neurons dropped during training; ReLU: rectified linear unit activation function.

**Table 3 sensors-25-01471-t003:** Method comparison results for the average emotion classification on the DEAP database.

Approach	Accuracy (%)	Kappa (%)	AUC (%)
EEGNet Valance	74.2±13.1	48.9±26.6	74.3±13.3
EEGnet Arousal	75.8±15.1	52.9±30.0	75.8±15.1
*Average*	75.0±14.1	50.9±28.3	75.1±14.2
EEGNet multiclass [53]	66.1±19.4	58.0±25.0	77.8±25.0
TCFussionnet multiclass [62]	57.7±15.3	45.7±19.8	71.3±10.3

**Table 4 sensors-25-01471-t004:** Performance evaluation metrics across comparisons.

Comparison	Metric	Mean	Mean Std Dev	Mean KL	Mean Overlap
Training vs. Training	*PC*	0.026631	0.024230	0.0	1.0
	*PR*	0.018504	0.010019	0.0	1.0
	*IOI*	3.184134	1.784040	0.0	1.0
	*PCH*	1.269862	0.963146	0.0	1.0
Training vs. Worst Subject	*PC*	0.042617	0.024720	0.446502	0.635858
	*PR*	0.026233	0.017207	0.207528	0.776660
	*IOI*	0.786089	0.728793	0.228678	0.751985
	*PCH*	1.101409	0.637367	0.236998	0.713150
Training vs. Best Subject	*PC*	0.099128	0.104696	0.082781	0.841051
	*PR*	0.082860	0.074627	0.249435	0.721505
	*IOI*	4.960142	6.567809	0.516072	0.676681
	*PCH*	0.994342	0.588620	0.965759	0.486928

## Data Availability

The datasets and Python 3.10 codes utilized in this study will be made available upon request via email.

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
