# Peer review of "EEG-Based Music Emotion Prediction Using Supervised Feature Extraction for MIDI Generation"

_sensors, 2025, doi:10.3390/s25051471_

Round 1
Reviewer 1 Report
Comments and Suggestions for Authors
The authors present a novel approach that integrates EEG-based emotion recognition with supervised feature extraction for music generation. The method is innovative in its use of deep learning architectures (e.g., EEGNet and autoencoder-based piano roll representations) to bridge the neural-auditory domains. This is an area of growing interest, particularly for applications in affective computing, music therapy, and brain-computer interfaces (BCI).
This reviewer suggests to frame the scientific contribution more explicitly. For example, the introduction describes challenges and related research, but does not explain well what is the research gap that this paper aims to address.
The study appears technical and application-driven, with limited theoretical discussion about how EEG-based emotion recognition contributes to the broader field of affective computing.
While the Abstract is very good, it could highlight the key contributions and how they advance the field. The problem statement is implicit. This makes it unclear to the reader why their approach is better over other models. Terms like "Centered Kernel Alignment" and "autoencoder-based piano roll representation" should be briefly explained in a way accessible to non-specialists.
The references used are appropriate covering relevant works on EEG-based emotion recognition; seep learning for music generation; and challenges in neural-music alignment. One gap, however, is the omission of references to studies that explore emotion recognition using alternative methodologies, such as facial expressions or multimodal emotion recognition. For instance, the paper would benefit from citing works such as Haddad & Laouris (2011: The ability of children with mild learning disabilities to encode emotions through facial expressions), which I have knowledge of (but there might be many others) evaluates facial expression-based emotion recognition in children with learning disabilities. This, and other studies that assess emotion perception using physiological signals such as skin conductance, facial electromyography, or heart rate variability, could offer a comparative perspective.
Along the same lines, the theoretical underpinnings of emotion perception and representation in the brain could be explored. There is limited discussion on how the proposed framework aligns with neurophysiological models of emotion perception (e.g., limbic system involvement in music-induced emotions). The relationship between EEG feature extraction and psychological theories of music perception could be elaborated on. Basically, other studies can be referenced and discussed to contrast EEG signals vs. facial expressions in decoding emotion and briefly discuss whether EEG-based music emotion recognition performs better than facial expression-based approaches. The paper could possibly acknowledge the benefits of combining EEG with other modalities (e.g., facial expressions, HRV, EMG). Making the above improvements could also suggest that future studies should explore fusion techniques for EEG and facial emotion recognition.
The authors could talk about possible shortcomings of their strategy. For instance, they could discuss the role of individual variability in EEG responses.
Furthermore, the generality of their framework is somewhat overstated. The results are promising, but their applicability to real-world music generation remains unclear. Claims about high-quality MIDI outputs should be qualified by addressing potential limitations (e.g., generalization across subjects, variability in EEG responses).
The Results are well-structured. A comparative analysis with alternative emotion recognition methods (vide supra) would add value.
The manuscript is strong and has significant potential. With some minor revisions (see below) to improve its theoretical grounding, discussion of limitations, and consideration of alternative emotion recognition approaches, the paper will be much stronger and better aligned with contemporary research trends in EEG-based emotion recognition and music generation.
Author Response
See attached pdf

Reviewer 2 Report
Comments and Suggestions for Authors
1. It’s an interesting issue, but I am really concerning the core processing steps are mostly cited from other existed references, which could lead no technique innovation in this paper.
2. The abstract is too long, and need to highlight the innovation and experiment performance in detailed.
3. The introduction description is too wide and not focus on this paper’s main propose, mostly are conceptual introduction not focus on the core problems, especially for this paper’s technique filed.
4. The core processing contain 5 steps, which are mostly cited from other existed papers. These formulars are also separately and not a unify or connect with each other by any parameters, that means these processing could not be calculated by one pass. If so, how to test this whole processing?
5. There contains many kinds of features, such as EEGNet-based Feature Extraction, Pianno Roll Features, auditory and Neural Features, et al, whether these features are the same data structure, or whether these features could cross-calculated with each other? These details were not given and no definitions. Even more for the “Emotion Label Prediction” and the calculation is not a prediction process, what kinds of emotions could be calculated and how many kinds?
6. Without giving the testing dataset size and scale, only introduced the channels and electrodes are not enough.
7. The Figure 2 is very strange, how to classify these results as shown in (b) and (c), that means these tests are almost cover the most emotions. Moreover, what’s the meaning of (a) shows, how to explain the “Emotion labels”?
8. I need to say, deep learning could not solve all kinds of classifications problem in high accuracy and performance requirements, especially for multi-kinds of classifications and without knowing the real features they are. So, why and how to use “deep learning framework” to solve the “k-nearest neighbor representations”?
9. The framework processing is mostly different with experiment aspects, they are focus on separately issues, what’s the final results? In the title, “Music Emotion Prediction”, how many kinds of music emotions were identified or could predict?
Author Response
See attached pdf
